# D3: A Dataset for Training Code LMs to Act Diff-by-Diff

**Ulyana Piterbarg**[*]
New York University

**Kanishk Gandhi**
Stanford University

**Lerrel Pinto**
New York University

**Noah Goodman**[†]
Stanford University

**Rob Fergus**[†]
New York University

## Abstract

We introduce D3 ("Diverse Data for Diff-by-Diff Coding"), a large dataset for training LMs to iteratively synthesize general-purpose Python source code by generating file diffs. D3 frames code synthesis as a goal-conditioned sequential decision-making problem, where goals, states, and actions are represented by token sequences corresponding to the description of a functionality to add, the current contents of a file, and a file diff, respectively. The dataset contains 8 billion tokens of instruction + file-state + file-diff-sequence examples sampled from 850,000 human-written Python source files. To construct D3, we filter, augment, and annotate source code from The Stack by sampling synthetic & linter-error-free file-diff sequences and labeling each sample with an LLM-generated rationale. In our experiments, we show that mid-training LMs like Llama 3.2 1b and 3b on D3 prior to supervised fine-tuning (SFT) on task-curated data improves performance on synthesis & editing tasks. On benchmarks like HumanEvalSynth and HumanEvalFix, we observe improvements in pass@1 of 3 to 6 points compared to direct SFT. D3-trained models are particularly strong at completing partial human-written solutions to programming problems.

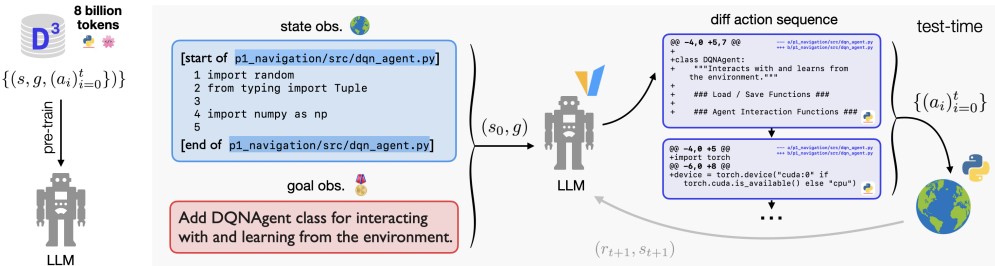

Figure 1: **The D3 dataset**. Diff action traces are checked for syntactical correctness with a Python analysis tool and capture a wide range of abstractions, from fine-grained additions to files like library imports to high-level architectural changes. The procedure used to prepare D3 is not specific to Python and can be used to create similar datasets for other languages.

## 1 Introduction

Achieving human-level performance in software engineering — let alone surpassing it — requires language models (LMs) to exhibit open-ended programming capabilities, i.e, be able to efficiently explore in program space and continuously add to or evolve the functionality and structure of a source file. While recent advances have enabled models to generate code to solve isolated programming problems from natural language instructions (Jaech et al., 2024; Dubey et al., 2024; Team et al., 2024), LMs and LM agents still fall short of replicating the iterative and exploratory nature of human software development (Jimenez et al., 2023; Yang et al., 2024; Pan et al., 2024; Antoniades et al., 2024).

---

[*]Corresponding author: up2021@nyu.edu. [†]Equal advising contribution

Code and the full D3 dataset will be made available at: https://github.com/upiterbarg/d3.

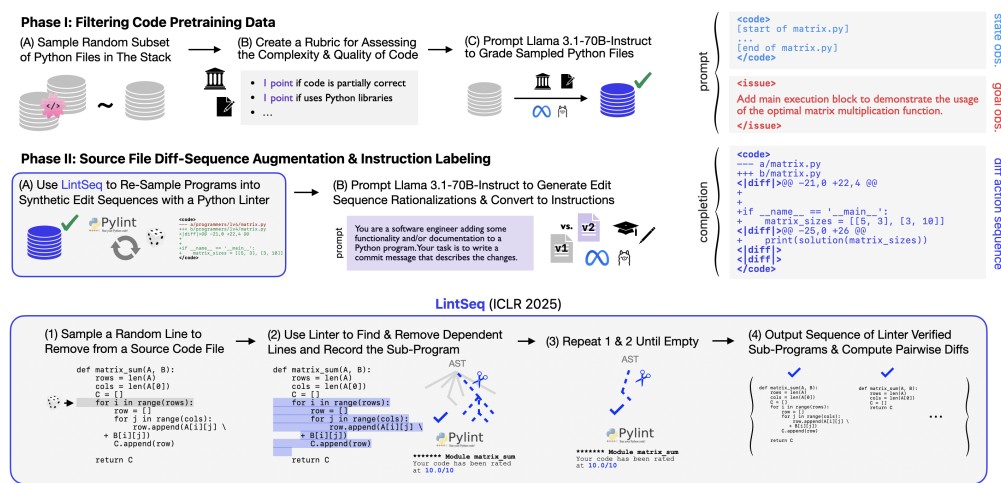

Figure 2: **(Top left) The procedure used to prepare D3**. There are two phases: source file filtering, and diff sequence sampling & instruction labeling. The first phase uses a LLM-judge and a data grading rubric (Appendix A, Figure 7), while the second uses an LLM-labeler and the LintSeq algorithm for sampling file-diffs. **(Top right) An instruction + file-state + file-diff-sequence example in D3**. **(Bottom) The LintSeq algorithm** for sampling synthetic file-diffs that reflect the semantics of a programming language.

One explanation for why models struggle with iterative code refinement (Olausson et al., 2023) is that they are predominantly trained on complete programs rather than edit data (Lozhkov et al., 2024b). Moreover, existing open-source datasets for coding lack: 1) scale & diversity — many datasets are optimized for downstream success on a narrow set of Python problem-solving datasets like MBPP and LeetCode (Wei et al., 2024), producing brittle models; 2) descriptive labels for edits — code edits are not necessarily paired with meaningful goal annotations (Muennighoff et al., 2023); and 3) human-like abstraction patterns, which may be helpful for iteratively exploring and designing code.

In an effort to address these limitations, we introduce D3 ("**D**iverse Data for **D**iff-by-**D**iff Coding"), a dataset of 3.6 million examples of instruction-annotated code edit sequences. D3 contains 8 billion tokens of data (Table 1), prepared through a novel combination of synthetic diff generation and LLM-powered filtering & labeling over permissively-licensed source code files from The Stack (Kocetkov et al., 2022). Our approach leverages the LintSeq algorithm to generate synthetic diffs that reflect programming language semantics and syntax (Piterbarg et al., 2024), a pretraining corpus filtering procedure inspired by FineMath (Lozhkov et al., 2024a), and an LLM-powered instruction labeling procedure (Wang et al., 2021) (see Figure 2). The data in D3 captures a wide range of programming patterns and abstractions, from low-level syntactic modifications to high-level architectural changes to source files, each paired with natural language instructions that describe the intended transformation (see Figure 1). Qualitatively, examples in D3 cover diverse topics like unit testing, data visualization, file I/O, image processing, and deep learning (see Figure 3).

Previous work has shown that breaking down code generation across sequences of edits or file-diffs can improve the quality & diversity of LM sampled code (Piterbarg et al., 2024). This mirrors how chain-of-thought prompting (Nye et al., 2021; Wei et al., 2021) can help language models solve certain complex reasoning tasks through intermediate steps. In our own set of experiments, we demonstrate each of the items below.

1. **D3 improves the performance of small LMs on synthesis & editing tasks**: Mid-training Llama 3.2 1b and 3b on D3 prior to supervised fine-tuning can improve pass@1 on single-file coding benchmarks by 3 to 6 points (Figure 4, Table 2).

2. **Ablating file-diffs from D3 hurts performance**: In the mid-training stage, replacing D3 file-diff sequences in favor of file re-writes – i.e., swapping diff action prediction for file-state prediction – degrades performance on synthesis tasks (Figure 5).

3. **LLM-powered source code filtering produces a "seed" corpus with better validation fit to synthesis & editing**: Training Llama 3.2 1b on Python files filtered with our approach results in improved loss on with-held examples from MBPP, BigCodeBench, & SWE-Bench compared to training on a random sample (Figure 5).

4. **Reliance on LLMs does not hurt data quality**: A human study on the quality of synthetic components in the D3 data preparation pipeline suggests broad agreement between LLM-judge vs human Python-expert file grades and edit descriptions.

Our work establishes that training on large-scale edit sequence data can be beneficial for improving the programming capabilities of language models. We see D3 as a first step towards the development of datasets for introducing human-like software development abstractions to LMs in earlier stages of training, providing a foundation for future work on open-source, autonomous, and open-ended programming agents.

Table 1: **Comparing D3 to existing datasets for training language models on file-level code editing and/or synthesis from instructions**. Tokens reflect the Llama 3.2 tokenizer.

| Dataset Name | Diffs? | Use-Case | Source | Task(s) | Toks | Toks/Ex |
|---|---|---|---|---|---|---|
| Evol-Instruct (Luo et al., 2023) | ✗ | post-train | synthetic | problem solving | 28M | 400 |
| OSS-Instruct (Wei et al., 2024) | ✗ | post-train | synthetic | problem solving | 37M | 500 |
| InstructCoder (Li et al., 2023) | ✔ | post-train | synthetic | code editing | 29M | 270 |
| LintSeq-Instruct (Piterbarg et al., 2024) | ✔ | post-train | synthetic | problem solving | 184M | 412 |
| CommitPackFT (Muennighoff et al., 2023) | ✔ | post-train | human | code editing | 275M | 400 |
| **D3 (ours)** | ✔ | **mid-train** | human + synthetic | **software synthesis** | **8.0G** | **2200** |

## 2 Problem Formulation

We formalize code synthesis as a goal-directed decision-making problem where an agent generates code through a sequence of edits to satisfy a natural language specification, as illustrated in Figure 1. This problem can be described as an MDP $(\mathcal{S}, \mathcal{G}, \mathcal{A}, \mathcal{T}, \mathcal{R})$, where:

- The state space $\mathcal{S}$ consists of states $s_i$, which each represent the contents and metadata of a source file or a multi-file codebase at a time step $i$.

- The goal space $\mathcal{G}$ corresponds to the set of possible code synthesis goals expressed in natural language, such that each $g \in \mathcal{G}$ corresponds to an instruction (shown as "goal obs." in Figure 1, e.g., "Add DQNAgent class for interacting with and learning from the environment").

- The action space $\mathcal{A}$ consists of individual diffs (atomic edits) that can be applied to the current file state $s_i$. As shown in the "diff action sequence" in Figure 1, actions $a \in \mathcal{A}$ are individual edits (e.g., adding an import statement or a class definition), each consisting of multiple tokens, that sequentially build up code changes.

- The transition function $\mathcal{T} : \mathcal{S} \times \mathcal{A} \to \mathcal{S}$ is deterministic and corresponds to applying the diff specified by action $a_i$ to the current file state $s_i$ to yield $s_{i+1}$.

- The reward function $\mathcal{R} : \mathcal{S} \times \mathcal{G} \times \mathcal{A} \times \mathcal{S} \to \mathbb{R}$ provides feedback about whether the modifications are moving toward satisfying the goal specification. As shown in the "test-time" component of Figure 1, this could involve executing the generated code.

The D3 dataset consists only of (goal, file-state)-action-sequence trajectories $(s_0, g, (a_i)_{i=0}^t)$, without reward annotations. These trajectories **do** capture successful code modifications that have been synthetically generated and verified for syntactical correctness with a code analysis tool, but **do not** include explicit reward signals.

The MDP formulation above captures several key challenges in code synthesis with language models. First, the LM must engage in long-horizon reasoning and goal understanding in order to plan sequences of mutually coherent edits. Second, the action space of possible diffs is combinatorially large and highly structured, necessitating efficient exploration strategies. In section 3, we describe how we construct a dataset of high-quality diff action sequences demonstrating successful code synthesis according to this problem formulation.

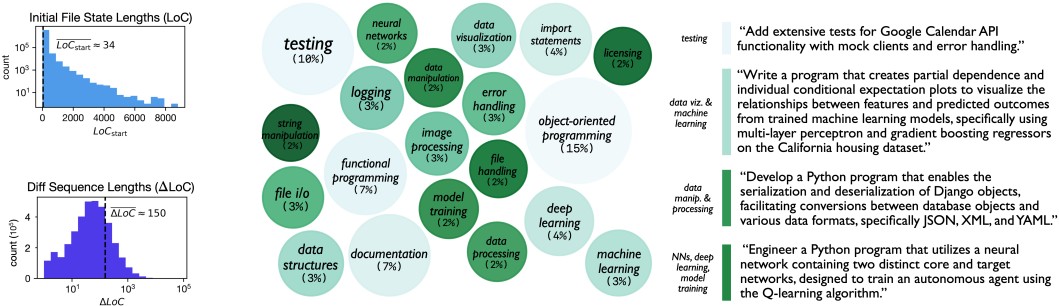

Figure 3: **Exploring the contents of D3**. (Left) Histograms for two metrics computed over the full D3 dataset: initial file state and diff sequence length in lines of code (LoC). (Right) Results of a qualitative topic discovery analysis run on instructions from D3 with Gemini ($n = 100{,}000$) (Team et al., 2024) (see Appendix D). We show the top $k = 20$ topics by incidence, and provide four labeled examples. Examples in D3 are qualitatively diverse.

## 3   D3: Diverse Data for Diff-by-Diff Coding

Our procedure for preparing D3 is largely unsupervised. In lieu of manual labeling, it relies on three components: a grading rubric for scoring source code for correctness, quality, & relevance to the study of software development in Python, an LLM capable of standard source code understanding, and the LintSeq sampling algorithm, which uses a linter to procedurally decompose source code files into atomic edit actions or line-by-line "chunks" (see Figure 2) (Piterbarg et al., 2024). By combining these tools, we show that it is possible to scalably re-purpose human-written code from the Web into instruction-labeled examples that are effective for training LMs to write and refine general-purpose programs. We also demonstrate that reliance on LLMs does not appear to impact data quality in a human study.

### 3.1   Preparation Procedure

An illustration of our method is provided in Figure 2. We prepare D3 by filtering Python files from The Stack, augmenting file contents into synthetic diff sequences, generating descriptions for each diff sequence, and converting descriptions into instructions.

**Filtering Code Pretraining Data with LLM-as-a-Judge**   Operating under computational constraints, we select a random sample representing about 10% of the total Python source code in the de-duplicated version of The Stack, consisting of 2 million unique files. Next, we filter this sample by prompting Llama 3.1 70b Instruct (Dubey et al., 2024) to assign a grade to each file. In each prompt, we provide the LLM with the full contents of a source file as well as a hand-engineered "grading rubric" that outlines a set of **four** correctness and content scoring criteria. These criteria draw from the math data-filtering prompt developed for the FineMath dataset by Lozhkov et al. (2024a), and are broadly designed to assess relevance to the study of software development in Python. The full prompt is provided in Appendix A. We truncate the contents of any file that exceeds 131,072 tokens.

Once all sampled source files have been graded, we use a simple parser to extract grades from each LLM output. We discard all files that were assigned a score lower than 3/4. This leaves us with 900,000 unique Python programs, representing $\approx 40\%$ of the initial file count.

**Augmentation and Labeling**   In this phase, our goals are two-fold: (1) to resample file contents into diff sequences; and (2) to label diff sequences with descriptive instructions. To accomplish the first of these objectives, we use the LintSeq algorithm and the popular Python linter pylint. As in Piterbarg et al. (2024), for each Python source file, we sample a sequence of intermediate sub-programs that represent a possible error-free *trajectory* for synthesizing the full file contents diff-by-diff. The complexity of each atomic diff action in this sequence is automatically determined by the size of a (syntactical) connected component in the underlying program graph (see Figure 2, bottom). Unlike Piterbarg et al. (2024), we

also sample four random *sub-trajectories* from each edit sequence, yielding some examples that reflect program synthesis "from-scratch" and others that reflect synthesis from partially written file contents. D3 reflects a 3:2 mix of these scenarios. In total, we add five instruction + file-state + file-diff-sequence examples per selected source file to the dataset.

Finally, we conclude our preparation of D3 by generating synthetic instruction labels for each edit sequence. Iterating over examples, we prompt Llama 3.1 70b Instruct to generate a complete but succinct ("lazy") rationale for the code that was added (see Appendix A). Then, we procedurally convert each rationale into an instruction, eliminating examples for which the LLM generated an incorrectly structured output. This labeling procedure allows us to circumvent the existing failure modes of current open-weight LLMs on diff understanding, while still leveraging their capabilities for cheap & scalable labeling.

## 3.2 What does the data in D3 look like?

As shown in Figure 3, examples in D3 span all scales of syntactical structures and abstractions in Python, from those that can be specified in just one line of code to others that requires thousands. The average example contains 34 lines of pre-written source code, and a corresponding diff sequence adding 150 lines of code. Distributions of both initial file lengths and diff sequence lengths are heavily right-skewed, with some examples having as many as 1000 lines of pre-written code. The content of examples in D3 also spans across a diverse range of topics, such as unit testing and deep learning (e.g., "Add tests for the Google Calendar API" or "Engineer a program to train an autonomous agent with Q-learning").

## 3.3 Assessing LLM-Generated Grades and Edit Descriptions with a Human Study

There are two LLM-controlled components that are critical to the preparation of D3: (1) source file grading, via a rubric and LLM-as-a judge; and (2) edit sequence labeling with "lazy" rationales, i.e. single-sentences descriptions of the changes made to code. Both components employ Llama 3.1 70b Instruct. To assess whether there is parity between LLM-generated versus human-annotated grades and labels, we conduct a human study.

### 3.3.1 Human Study Design

In Part A, four human annotators with expertise in Python were presented with the same rubric provided to the LLM judge and asked to provide scores and per-item confidence ratings for 24 randomly selected Python files from the sample of The Stack used to prepare D3. Human consensus grades for each file were computed as the arithmetic mean, median, and confidence-weighted mean of the four human scorers.

In Part B, the same set of annotators were provided with "overall" diffs corresponding to the starting vs final files states for 20 randomly selected samples from D3. Each diff was paired with two (order-shuffled) descriptions: the LLM-generated description for the sample present in D3 and a human-written description, prepared by a fifth study participant who was asked to independently label the sample via the same set of instructions given to the LLM-labeler. Annotators were then instructed to indicate their preferred "higher-quality" description single-blind, and to provide a confidence. For each diff, the number of votes for the LLM-generated description (0–4) was tallied, majority was defined as $\geq 3/4$ votes, and 2–2 ties were retained for agreement summaries but excluded from the sign test.

### 3.3.2 Human Study Results

Across the 24 files in Part A, we find that Llama 3.1 70b scores closely track human consensus with no systematic score shift: the median difference and Hodges–Lehmann paired shift for LLM vs human scores were 0.00 against all three human consensuses definitions, mean, median, confidence-weighted mean (Hodges & Lehmann, 1963). Furthermore, paired Wilcoxon tests were non-significant ($p = 0.54, 0.50, 0.54$, respectively), absolute deviations were small (MAE $\approx$ 0.49–0.50), and rank-order agreement was strong (Spearman $\rho =$ 0.78–0.84, Kendall $\tau - \beta =$ 0.71–0.74, all $p < 0.01$) (Wilcoxon, 1945; Spearman, 1904; Kendall,

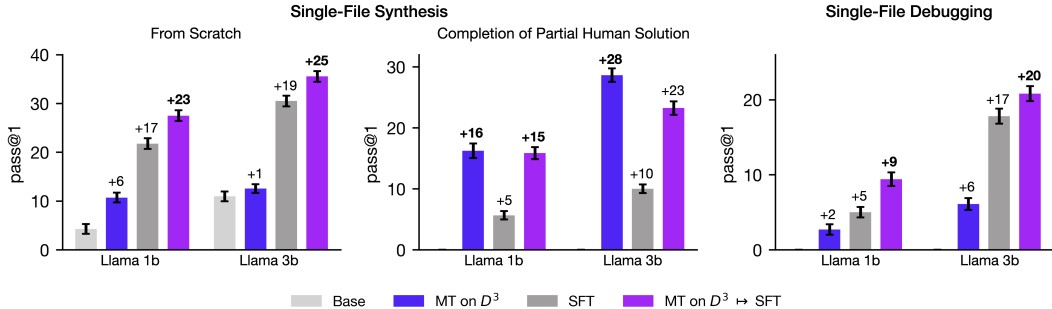

Figure 4: **Task-aggregated pass@1 on single-file synthesis & debugging** for base, mid-trained (MT), and supervised fine-tuned Llama 3.2 1b and 3b models (temperature = 0.1). For a breakdown of results by benchmark, see Table 2 below. Base models are evaluated with the D3 prompt-completion structure (see Figure 2). A description of evaluation procedures and examples of formatted prompts are provided in Appendix I.

1945). Overall, the LLM-judge scores used for source code filtering in the first phase of D3 preparation are aligned and unbiased relative to human judgments.

On the 20 diffs in Part B, the LLM-generated description was the majority choice for 8/20 items (40%, exact 95% CI 0.19–0.64). A sign test on non-tied items (8 wins vs 6 losses; $n = 14$) found no evidence of a departure from parity (two-sided $p = 0.79$). Vote dispersion per item between annotators was moderate to high (mean Shannon entropy 0.71 bits; 0 = unanimity, 1 = 2–2 split), and chance-corrected agreement among raters was slight (observed agreement $A_0 = 0.55$ and Fleiss' $\kappa = 0.10$), consistent with meaningful ambiguity and spread in preferences (Fleiss, 1971). At this sample size, we conclude that the data do not indicate a clear overall annotator preference for either LLM-generated or human-written labels.

In summary, **our human study affirms the quality of synthetic components in D3** – our data suggest broad parity between LLM and Python-expert file grades, and no significant deviation in quality between LLM-generated and human-written edit descriptions.

# 4 Mid-training and SFT Experiments

We further study the effectiveness of D3 by conducting a series of mid-training and SFT experiments with the Llama model series. These experiments seek to answer the following:

1. Does our filtering procedure produce a corpus that yields better fit on tasks like problem-solving & real-world editing, compared to a random sample of programs?

2. How does ablating diff-by-diff generation from D3 affect performance? In other words, does swapping diff actions for full file-state prediction from demonstration data affect LM generation quality at test-time?

3. Does training language models on D3 before task-specific fine-tuning improve their performance on single file and repository-wide coding tasks?

## 4.1 Ablation Study: Testing the importance of filtering and diff-by-diff generation

**Filtering**   First, to test the effectiveness of our LLM-assisted data filtering procedure, we continue to train the pretrained model weights on corpuses of filtered vs unfiltered Python source code. Unfiltered files are drawn from the sample of graded data (see subsection 3.1) and randomly sampled to match the token count of filtered code. Throughout each experiment, we estimate LM accuracy on problem solving & real-world code editing tasks by computing average loss on a dataset of withheld prompt-completion examples drawn from MBPP, BigCodeBench-Hard, and SWE-Bench (Oracle) (Austin et al., 2021; Zhuo et al., 2024; Yang et al., 2024). Prompt tokens are masked out.

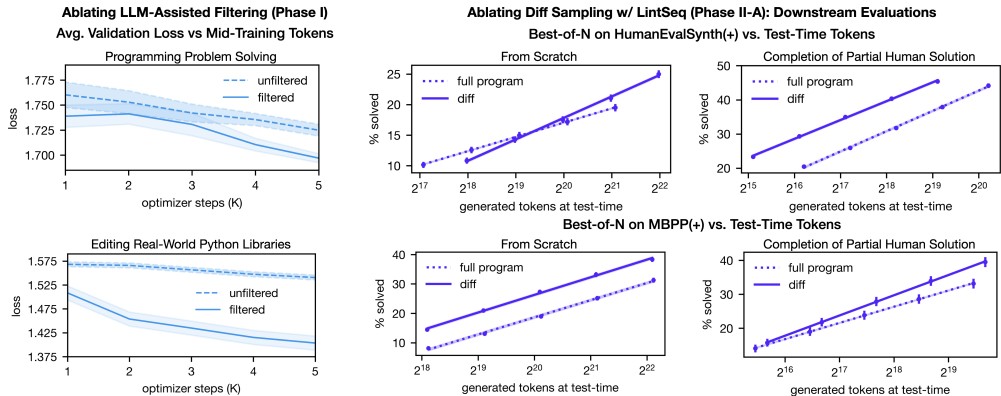

Figure 5: **(Left) Data filtering ablation**: smoothed loss on withheld examples by task while mid-training Llama 3.2 1b on The Stack, with and without the filtering procedure described in Section 3.1. Shading indicates value ranges before smoothing. Filtering improves loss both on Python problem solving and real-world library editing examples. **(Right) File-diff-sequence ablation**: tuned test-time scaling on HumanEvalSynth and MBPP for Llama 3.2 1b models mid-trained on D3 with and without file-diff-sequences. Models trained to synthesize code with diffs exhibit better best-of-n scaling and pass@1 (Table 6, Table 7).

**Diff-by-Diff Generation**  Next, we probe the effect of training LMs to synthesize source code diff-by-diff by mid-training Llama 3.2 1b on the D3 dataset "as is," as well as on a version of the dataset where file-diff sequences are replaced with full file states in each example. We train models on both corpora for two epochs and with identical hyperparameters. To compare models, we evaluate the test-time performance of intermediate checkpoints on the code synthesis benchmarks HumanEvalSynth and MBPP (Muennighoff et al., 2023; Austin et al., 2021; Liu et al., 2023). For each task, we test two settings: (A) **synthesis "from scratch"**, where models are prompted to write a program according to an instruction starting from an empty file-state; and (B) **synthesis from a partially-written program**, where models are prompted to complete a partial human solution to each problem. Setting (B) is procedurally generated with a fixed random seed from existing test set solutions (see Appendix I).

**Ablation Results**  We provide visualizations of results in Figure 5. As shown in Figure 5 (left), filtering improves accuracy on both problem solving & real-world code editing examples throughout training. This suggests that despite the potential biases introduced by our LLM-assisted data labeling procedure, the "seed" corpus of source code that we augment and annotate to produce D3 yields a better fit to examples of tasks of interest compared to a random sample of source files with similar token count. The degree of improvement on real-world editing examples especially suggests that filtering balances quality against diversity. Furthermore, as shown in Figure 5 (right), models mid-trained to synthesize code diff-by-diff on D3 exhibit better test-time scaling laws compared to their full program regeneration counterparts, reinforcing the conclusions of Piterbarg et al. (2024).

## 4.2 Following Mid-training on D3 with Task-Targeted SFT

Next, we test whether there are downstream benefits to mid-training 1b and 3b Llama models on D3 prior to supervised-finetuning (SFT) on task-curated instruction data. As described in Section 3, D3 consists of diverse & high-quality examples of general-purpose software synthesis in Python; consequently, we expect task-curated SFT to be beneficial for improving the precision of D3-mid-trained model generations on problem solving & debugging tasks in benchmarks like HumanEvalSynth and HumanEvalFix.

To that end, we first mid-train 1b and 3b models on D3 for two complete epochs. Then, we run a series of SFT experiments in which we tune the base and D3-mid-trained models on one of three data variants, curated by task: **synthesis/problem-solving**, consisting of instruction + file-diff-sequence examples from LintSeq-Instruct (Piterbarg et al., 2024);

| Size | MT on D3? | SFT? | HumanEvalSynth(+) | | MBPP(+) | | HumanEvalFix(+) |
| | | | Scratch | Compl. | Scratch | Compl. | Debugging |
|---|---|---|---|---|---|---|---|
| 1b | ✗ | ✗ | $5.5 \pm 1.0$ | $0.0 \pm 0.0$ | $3.0 \pm 1.0$ | $0.0 \pm 0.0$ | $0.0 \pm 0.0$ |
| 1b | ✔ | ✗ | $9.2 \pm 1.0$ | $\mathbf{21.0 \pm 1.2}$ | $12.2 \pm 1.0$ | $11.5 \pm 1.0$ | $2.7 \pm 0.7$ |
| 1b | ✗ | ✔ | $18.0 \pm 1.1$ | $3.6 \pm 0.7$ | $25.5 \pm 1.1$ | $7.7 \pm 1.0$ | $5.0 \pm 0.7$ |
| 1b | ✔ | ✔ | $\mathbf{23.1 \pm 1.1}$ | $16.1 \pm 1.0$ | $\mathbf{31.9 \pm 1.1}$ | $\mathbf{15.6 \pm 1.1}$ | $\mathbf{9.4 \pm 0.9}$ |
| 3b | ✗ | ✗ | $8.5 \pm 1.0$ | $0.0 \pm 0.0$ | $13.4 \pm 1.0$ | $0.0 \pm 0.0$ | $0.0 \pm 0.0$ |
| 3b | ✔ | ✗ | $11.0 \pm 0.9$ | $\mathbf{35.7 \pm 1.1}$ | $14.1 \pm 0.8$ | $21.6 \pm 1.1$ | $6.1 \pm 0.8$ |
| 3b | ✗ | ✔ | $27.0 \pm 1.1$ | $5.8 \pm 0.7$ | $34.0 \pm 1.2$ | $14.2 \pm 1.0$ | $17.8 \pm 1.0$ |
| 3b | ✔ | ✔ | $\mathbf{30.9 \pm 1.1}$ | $27.1 \pm 1.1$ | $\mathbf{40.2 \pm 1.2}$ | $\mathbf{19.4 \pm 1.2}$ | $\mathbf{20.8 \pm 1.0}$ |

Table 2: **Full pass@1 results on single-file synthesis & debugging by benchmark** at temperature $= 0.1$ ("$\pm$" indicates standard error over $n = 20$ samples).

| Model | Size | D3? | SFT? | % Resolved | | % Apply | |
| | | | | pass@1 | pass@10 | pass@1 | pass@10 |
|---|---|---|---|---|---|---|---|
| Ours | 1b | ✗ | ✔ | $0.56 \pm 0.15$ | $1.78 \pm 0.36$ | $\mathbf{36.33 \pm 0.60}$ | $93.73 \pm 0.48$ |
| Ours | 1b | ✔ | ✔ | $\mathbf{0.92 \pm 0.18}$ | $\mathbf{3.00 \pm 0.45}$ | $36.02 \pm 0.80$ | $\mathbf{87.85 \pm 0.70}$ |
| Ours | 3b | ✗ | ✔ | $1.15 \pm 0.19$ | $3.43 \pm 0.51$ | $55.35 \pm 0.81$ | $97.22 \pm 0.33$ |
| Ours | 3b | ✔ | ✔ | $\mathbf{1.71 \pm 0.32}$ | $\mathbf{4.81 \pm 0.56}$ | $\mathbf{69.60 \pm 0.81}$ | $\mathbf{99.62 \pm 0.17}$ |
| SWE-Llama(†) | 7b | - | - | $3.01 \pm \text{??}$ | - | $66.78 \pm \text{??}$ | - |

Table 3: **Scaffold-free results on SWE-Bench (Oracle)** at temperature $= 0.5$ ("$\pm$" indicates standard error over $n = 16$ samples). "% Resolved" denotes the percentage of GitHub issues correctly solved by the model, while "% Apply" denotes the percentage of generated patch sequences that contain only well-formed file-diffs. "†" indicates scores drawn from Jimenez et al. (2023), reflecting performance after in-domain SFT on the train split of SWE-Bench.

**editing**, consisting of Python file-state + instruction + file-diff examples from CommitPackFT (Muennighoff et al., 2023); and lastly, **synthesis/problem-solving + editing**, reflecting a mix of examples from both LintSeq-Instruct & CommitPackFT (see Appendix F and Appendix G).

Our experimental results are shown in Figure 4, Table 2, and Table 3. In aggregate, we evaluate all models on six tasks spanning four benchmarks, HumanEvalSynth, MBPP, HumanEvalFix, and SWE-Bench (Oracle) (Austin et al., 2021; Muennighoff et al., 2023; Liu et al., 2023). For each setting and model variant, we report scores from the best performing SFT mix only. As in subsection 4.1, we evaluate LMs on two variants of each synthesis/problem-solving task in order to test model ability to both synthesize a solution to a programming problem from scratch (denoted as "Scratch" in Table 2), and to complete a partially-written human solution (denoted as "Compl.").

On single-file synthesis & editing, mid-training models on D3 prior to SFT improves pass@1 by margins of 3 to 6 points compared to direct SFT on base models (Figure 4, Table 2). Furthermore, we observe that on the completion variants of HumanEvalSynth and MBPP tasks, fine-tuning models mid-trained on D3 degrades performance over mid-training alone.

On the challenging repository-level code editing benchmark SWE-Bench (Oracle), we similarly find a statistically-significant gain from mid-training on D3 before SFT. It is important to note that this evaluation setting is somewhat out-of-distribution from both D3 and from all three of our tested SFT dataset variants in two respects: first, models may be provided the state of more than one file in-context; and second, many prompts span tens of thousands of tokens in length, exceeding our mid-training and SFT context lengths (4096). Even so, mid-training on D3 improves pass@10 score on "% Resolved" (the total proportion of solved GitHub issues in the test set) by a relative margin of 60% and 40% for 1b and 3b Llama models, respectively. It also especially improves "% Apply" (the fidelity of file-diffs in generations) for the 1b model, improving raw pass@10 by 6 points.

In summary, the results of our dual-stage training experiments with D3 suggest that mid-training small language models on the dataset is indeed effective for improving downstream performance across coding tasks. D3 appears to be especially effective for improving the quality of generations on completion-like synthesis tasks, where models must add code to partially pre-written programs.

## 5    Related Work

**Language Models for Code Generation**    Modern language models are pretrained on ter-abytes of code. Programming benchmarks like HumanEvalSynth (Muennighoff et al., 2023) and MBPP (Austin et al., 2021) evaluate models on their ability to solve simple "homework-like" code synthesis problems and constitute a core part of generative LM evaluations (Dubey et al., 2024; Achiam et al., 2023). Recently, there has also been increasing interest in training and testing LMs on more realistic coding tasks. For example, benchmarks like HumanEvalFix (Muennighoff et al., 2023) and SWE-Bench (Jimenez et al., 2023) evaluate generative code LMs on program- and repository-level code editing. While these benchmarks have spurred improvement in LM capabilities, contemporary open-source models continue to lag behind their closed-source counterparts, particularly on the hardest of these tasks (Jimenez et al., 2023). Available open-source resources for synthesis like OSS-Instruct (Wei et al., 2024) and Evol-Instruct (Luo et al., 2023) focus on narrow single-function generation tasks (similar to HumanEval), while GitHub commit-derived datasets (Muennighoff et al., 2023) lack consistent instruction labels and may fail to capture the granular abstractions used by human engineers during software development. By open-sourcing the D3 dataset & preparation pipeline, we hope to contribute towards addressing these data limitations.

**Training on Code Edits**    Many existing works study data-driven code editing with language models. Berabi et al. (2021) re-label GitHub commit data with error messages generated by a code analysis tool, and train an encoder-decoder transformer on this data. Muennighoff et al. (2023) train LMs on filtered GitHub commit and message pairs, while Cassano et al. (2023) use filtered commit data to seed synthetic instruction data generation with ChatGPT/GPT-4. More recently, Piterbarg et al. (2024) study edit sequence representations for code synthesis. Their experiments show that fine-tuning on synthetic diffs improves test-time scaling laws for code synthesis across LM parameter scales. Our dataset builds directly upon this work, using the LintSeq algorithm for refactoring filtered, human-written source code files at pre-training scales.

**Language Models for Data Labeling & Filtering**    Since the release of ChatGPT/GPT-4, it has become fairly commonplace to use LLMs to generate and/or augment instruction data (Taori et al., 2023; Chung et al., 2024; Wang et al., 2023b; Chiang et al., 2023). Such methods are closely related to the literature on knowledge distillation and self-improvement (Hinton, 2015; Kim & Rush, 2016). More recently, several works have also explored leveraging LMs for data filtering. Penedo et al. (2024); Lozhkov et al. (2024a) show that prompting strong instruction-tuned models to grade Web-text in pretraining corpora for quality and relevance can result in boosts to downstream performance on domains like math reasoning. Relatedly, Li et al. (2024) introduce Superfiltering, showing that LM-assisted filtering can result in models that outperform the labeler on the downstream task-of-interest, unlike distillation. Our procedure for preparing D3 is motivated by these works, and the potential of LM filtering & labeling methods for producing code datasets that balance quality and diversity with limited human labor and relatively low cost.

## 6    Discussion

We introduced D3, a dataset of 3.6 million instruction-annotated code edits derived from 850,000 human-written and highly-diverse Python programs. The dataset's construction leveraged three key components: a pretrained LLM for filtering, the LintSeq algorithm (Piterbarg et al., 2024) for synthetic edit generation, and an instruction annotation pipeline. This comprehensive approach has yielded a dataset with several distinguishing charac-

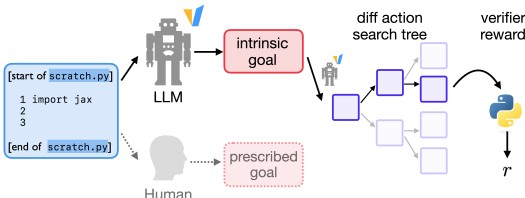

Figure 6: **Towards training LMs for open-ended, goal-directed, & autonomous software development by leveraging D3**. Future work might explore following up mid-training on D3-like data with RL or test-time search over self-generated goals and/or diff action sequences. Our data preparation pipeline could also be used for further study of data-driven methods for improving the exploration capabilities of LMs during generation, beyond D3.

teristics - notably its coverage of both complete development scenarios (60%) and partial file modifications (40%), along with substantially larger context windows averaging 2,200 tokens compared to the typical 400-500 tokens of existing instruction data.

Our experimental results validate the quality of LLM-generated components and the effectiveness of training on diverse edit sequences generated from Human-written Python source code files. Mid-training LMs on D3 prior to task-targeted SFT produced significant benefits across multiple tasks. These improvements were particularly pronounced in refinement-style synthesis settings, where success hinges on deep understanding of code context and development patterns. This suggests that our edit-based approach captures fundamental aspects of software development that existing synthesis instruction datasets miss.

**Limitations**   While these results are promising, several important limitations warrant discussion. Our current focus on Python, though practical for initial development, raises questions about generalizability to other programming languages with different syntactic structures and development patterns. Additionally, despite the empirical results from our human study and training experiments, our reliance on LLM-assisted filtering & instruction labeling in the creation of D3 may introduce systematic biases and coverage gaps. Furthermore, our procedure for generating instruction labels for edit samples post-hoc may introduce systematic goal under-specification into the dataset.

**Future Work**   Looking forward, we see the use of diffs as a fundamental representation opening up compelling possibilities for open-ended software development, as well as for both in-context (Gandhi et al., 2024) and external search (Schultz et al., 2024) in code generation, as shown in Figure 6. Diffs provide an ideal action space for search, and are particularly well-suited for exploring program space incrementally. This representation allows models to decompose complex programming tasks into atomic line-by-line actions that are guided by syntax while incorporating feedback at each iteration – an approach that echoes agile software development practices used by human engineers (Dybå & Dingsøyr, 2008). Future work could explore combining this diff-based representation with reasoning and RL (Gehring et al., 2024) to create more adaptive & context-aware systems.

These future directions, combined with our current results, suggest that edit-based approaches to code generation represent a promising path toward more capable and practical code generation systems. The success of D3 demonstrates the fundamental value of modeling software development as an iterative, goal-driven process at the data-level, rather than treating it as a single-step and purely state-prediction-based generative task.

## Acknowledgments

This work was supported by NSF award 2339096 and ONR awards N00014-21-1-2758 and N00014-22-1-2773. UP is funded by an NSF GRFP award, KG is funded by an HAI-SAP Grant and an NSF Expeditions grant, and LP is funded by the Packard Fellowship. Experiments were conducted with compute from Stanford Marlowe and NYU Greene HPC. We are also grateful to the Gemini Academic Program and Divy Thakkar for their support.

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

# A    Dataset Preparation Prompts

> Evaluate the following Python code extract for its potential usefulness for studying Python programming up to the competitive programming level.
>
> Use the following **4-point scoring system** described below. Points are accumulated based on the satisfaction of each criterion:
>
> - Add 1 point if the extract **contains some correct code that reflects foundational Python programming concepts**, even if its not very useful for solving hard programming problems.
> - Award another point if the extract **correctly demonstrates examples of how to use common Python libraries**.
> - Add a third point if the code correctly uses a **more advanced data structure or algorithm**.
> - Award a fourth point if the extract reflects Python code that is **outstanding in its educational value for competitive programming**.
>
> The Python code extract:
> ```
>
> {program}
> ```
>
> After examining the extract:
> - Briefly justify your total score, up to 50 words.
> - Conclude with the score using the format: Final score: <total points>.

Figure 7: The grading rubric we use for **prompting Llama 3.1 70B Instruct to score the quality and relevance of Python source files contents** for the study of high-quality software synthesis, used in the filtering stage of our D3 preparation procedure. The structure and design of this prompt is inspired by work from Lozhkov et al. (2024a) on LLM-powered filtration of pretraining data for math reasoning.

> You are a software engineer adding some functionality and/or documentation to a (possibly empty) Python program. Your task is to write a **commit message** that describes the changes that you made that is:
>
> 1. In imperative tense
> 2. Succinct, clear, and describes the changes exactly
> 3. No longer than one sentence
> 4. Does not use first person pronouns
> 5. Professionally written
>
> The original Python program:
> ```
>
> {program_start}
> ```
>
> The Python program after you made changes to it:
> ```
>
> {program_end}
> ```
>
> After comparing the Python programs above, provide your commit message describing the functionality and/or documentation that was added using the format: Commit message: <commit message>.

Figure 8: Prompt for **generating rationales for examples of partial source file synthesis** with Llama 3.1 70B. We replace fields annotated as {program_start} and {program_end} above with the contents of the source file before and after all atomic diff actions are applied, respectively.

You are a software engineer. Describe the Python program below using a single sentence that is:
1. In imperative tense
2. Succinct, clear, and describe the changes exactly
3. Free of first person pronouns
4. Professionally written

The Python program:
```

{program}
```

After examining the program, provide your single sentence description using the format: This is a Python program that: <my description>.

Figure 9: Prompt for **generating rationales for examples of full-file synthesis "from scratch"** with Llama 3.1 70B. We replace the {program} field with the final file-state.

# B  Full-Text D3 Examples

```

[start of programmers/lv4/matrix.py]
  1 # 최적의 행렬
  2
  3 INF = 987654321
  4
  5
  6 def solution(matrix_sizes):
  7     dp = [[INF for _ in range(len(matrix_sizes))] for _ in range(len(matrix_sizes))]
  8     solve(0, len(matrix_sizes) - 1, dp, matrix_sizes)
  9     return dp[0][len(matrix_sizes) - 1]
 10
 11
 12 def solve(y, x, dp, matrix):
 13     if y == x:
 14         dp[y][x] = 0
 15         return dp[y][x]
 16
 17     if y + 1 == x:
 18         dp[y][x] = matrix[y][0] * matrix[y][1] * matrix[x][1]
 19         return dp[y][x]
 20
 21     return dp[y][x]
[end of programmers/lv4/matrix.py]


<issue>
Add main execution block to demonstrate the usage of the optimal matrix multiplication function.
</issue>


--- a/programmers/lv4/matrix.py
+++ b/programmers/lv4/matrix.py

<|diff|>@@ -21,0 +22,4 @@
+
+
+if __name__ == '__main__':
+    matrix_sizes = [[5, 3], [3, 10], [10, 6]]
<|diff|>@@ -25,0 +26 @@
+    print(solution(matrix_sizes))
<|diff|>
<|diff|>

```

Figure 10: Full-text version of the D3 example shown in Figure 2. We train LMs to interleave atomic diff actions using the use the special token `<|diff|>` during generation.

```

[start of p1_navigation/src/dqn_agent.py]
   1 import random
   2 from typing import Tuple
   3
   4 import numpy as np
[end of p1_navigation/src/dqn_agent.py]


<issue>
Add DQNAgent class for interacting with and learning from the environment.
</issue>


--- a/p1_navigation/src/dqn_agent.py
+++ b/p1_navigation/src/dqn_agent.py

<|diff|>@@ -4,0 +5,9 @@
+
+
+
+class DQNAgent:
+    """Interacts with and learns from the environment."""
+
+    ### Load / Save Functions ###
+
+    ### Agent Interaction Functions ###
<|diff|>@@ -4,0 +5 @@
+import torch
@@ -6,0 +8 @@
+device = torch.device("cuda:0" if torch.cuda.is_available() else "cpu")
<|diff|>@@ -15,0 +16,8 @@
+
+    def learn(self, experiences, gamma):
+
+        states, actions, rewards, next_states, dones, idxs = experiences
+
+
+
+        # ------------------- update target network ------------------- #
<|diff|>@@ -5,0 +6 @@
+import torch.optim as optim
<|diff|>@@ -17,0 +18,5 @@
+    def act(self, state, eps=0.):
+
+        state = torch.from_numpy(state).float().unsqueeze(0).to(device)
+
+        # Epsilon-greedy action selection
<|diff|>@@ -12,0 +13,30 @@
+
+    def __init__(
+            self,
+            state_size,
+            action_size,
+            update_type  = 'dqn',
+            LR           = 1e-3,       # 1e-3 has optimal training times
+            TAU          = 1e-3,       # for soft update of target parameters
+            BUFFER_SIZE  = int(1e5),   # 88Mb memory + last 100 games @ 1000 timesteps per game
+            BATCH_SIZE   = 64,         # minibatch size - sensible default
+            UPDATE_EVERY = 4,          # how often to update the network
+            **kwargs
+    ):
+
+        self.BATCH_SIZE   = BATCH_SIZE
+        self.BUFFER_SIZE  = BUFFER_SIZE
+        self.TAU          = TAU
+
+
+        self.state_size  = state_size
+        self.action_size = action_size
+        self.update_type = update_type
+
+        # Q-Network
+
+        # Replay memory
+        # NOTE: training using SumTreeReplayBuffer fails to converge
+
+        # Initialize time step (for updating every UPDATE_EVERY steps)
+
<|diff|>@@ -18,0 +19 @@
+            seed         = 42,
@@ -30,0 +32 @@
+        random.seed(seed)
<|diff|>@@ -0,0 +1 @@
+import os
@@ -47,0 +49,11 @@
+    def load(self, filename):
+        try:
+            filename = os.path.abspath(filename)
+        except Exception as e: print(f'{self.__class__.__name__}.load(): {filename} exception: {e}')
+
+
+    def save(self, filename):
+        filename = os.path.abspath(filename)
+
+
+
<|diff|>@@ -56,0 +57,3 @@
+        try:
+            print(f'\n{self.__class__.__name__}.save(): {filename} = {os.stat(filename).st_size/1024:.1f}kb')
+        except Exception as e: print(f'{ self.__class__.__name__}.save(): {filename} exception: {e}')
<|diff|>
<|diff|>

```

Figure 11: Full-text version of the D3 example shown in Figure 1.

## C  More Examples of Instructions in D3

Add pickle import to step_one function and specify pickle_url in executing_pipeline function call.

Add file with author information and metadata.

Implement argument parser to handle command-line arguments and add PyTorch model training functionality.

Add settings, data structures, and classes for CSC trainer functionality.

Implement HTML input stream functionality with encoding detection and error handling.

Develop a Python program that implements utility classes and functions to parse and resolve network addresses, encompassing IPv4 and IPv6 addresses, from strings, uniform resource identifiers (URIs), and sockets.

Implement initial Convolutional Neural Network structure for Van der Pol equation solution prediction.

Add argument -- num-doc-keys to parser_run and modify create_doc function to include num_doc_keys.

Add instance variables for stride and label_patch_radius in BoxDef class.

Add utility functions and skeleton for defining custom Matplotlib colormaps.

Implement str_to_class function, add Package class with subclassing and abstract methods, and skeleton classes for GalaxyPackage and ShellScriptPackage.

Add documentation for command line utility and extract version number from CVS tag.

Add the missing 'z' argument to the cli function in the Click command decorator.

Add XSRF token classes and functionality for generating and verifying token strings.

Develop a Python program that utilizes the Pytest framework to test the authorization mechanisms and API requests of a TAXII server application, with a focus on scenarios including unauthorized access, basic authentication, token-based authentication, and collection-level access control.

Add support for report-to, custodian, lifetime, and default outgoing flags in serialize_bundle6.

Add function to implement Trotterization for a give list of terms, ordering, time and Trotter number.

Update WaypointUpdater class to initialize and handle lane waypoints and provide placeholder methods for pose, waypoint, obstacle, and distance calculations.

Add required imports for struct and time modules.

Implement BertAttention, BertLayer, and BertEncoder modules with chunking and checkpointing functionality.

Add core imports, environment setup, and key variable definitions for data processing.

Update `build_vocab` method in `TransfoXLTokenizer` to provide size information of the final vocabulary and the number of unique tokens.

Add Connection class with attributes and methods for serializing to JSON and constructing from a list of fields.

Implement integration and gradient evaluation functions for vanishing and nonvanishing basis elements.

Develop a Python program that implements the Pepper compiler and a suite of tests to verify its build steps, leveraging mock objects to mimic actual file system interactions.

Develop a Python program that utilizes PyTorch Lightning to construct and train a multi-label classification system, then evaluate its performance on a provided medical image dataset.

Add initial response reading function and import wait coroutine from part 5.

Initialize Context class for register and flag manipulation.

Add docstrings to describe program output and data structure.

Add logging configuration to the test environment.

Using Python, engineer a program that leverages Bayesian optimization for hyperparameter tuning in a random forest classifier, ultimately employing the optimized model to generate predictions from a test dataset, with the results exported to a CSV file.

Implement lane detection image generation and accuracy calculation functionality for CULane and Caltech datasets.

Figure 12: A random sample of $n = 32$ source code development instructions in D3.

# D    Qualitative Analysis of Topics in D3

You are a coding expert. When I ask you to output data with JSON formatting, please return only correctly formatted JSON without an explanation.

Analyze the following Python coding instruction and identify:
1. Main topics/concepts
2. Programming libraries/frameworks mentioned or implied
3. Tools or applications involved
4. Level of complexity (beginner, intermediate, advanced)
5. Type of task (data analysis, web development, automation, etc.)

Return your analysis as a JSON object with the following fields:
- main_topics
- libraries_frameworks
- tools_applications
- all_complexity
- all_task_types

Instruction to analyze: {instruction}

Figure 13: Our prompt format for **qualitatively analyzing the contents of D3 via annotation of 100,000 randomly sampled instructions from the dataset with Gemini**. This prompt is effective for producing only valid `JSON` outputs from the model. To make the visualization of top occurring topics in section 3, Figure 3 (right), we parse the `main_topics` field of each output JSON, manually group paraphrased outputs, and sort by topic frequency.

# E    Human Study

## E.1    Participant Recruitment & Ethics

All annotators were volunteers who self-selected after viewing a short recruitment notice. During screening they affirmed at least intermediate proficiency in Python ($\geq$ 4 years of regular use) and provided electronic informed consent outlining study purpose, minimal risks, data handling, and optional withdrawal at any time. No personal identifiers beyond platform IDs were collected, and participation carried no mandatory compensation. Data reported in the manuscript corresponds only to responses from those volunteers who completed every task.

## F  Mid-Training

We use Pytorch FSDP and the Dolma tokenization toolkit via the `oLMo` ecosystem to support all mid-training experiments (Zhao et al., 2023; Soldaini et al., 2024; OLMo et al., 2024). We add a new token "`<|diff|>`" to tokenizer vocabularies, and resize model embeddings accordingly. This special token is used to separate individual file-diffs in example diff action trajectories, effectively indicating a frame "reset" in diff integration (see Figure 11, Figure 10, and Piterbarg et al. (2024)). Mid-training runs use two to four NVIDIA A100 or H100 GPUs.

|  | Llama 3.2 1b & 3b |
| --- | --- |
| Batch Loss Reduction | mean |
| Batch Size | 256 |
| Betas | $(0.9, 0.95)$ |
| Gradient Clipping | 1 |
| Flash-Attention | true |
| Learning Rate Scheduler | cosine |
| Max Learning Rate | 1e-4 |
| Max Sequence Length | 4096 |
| Mixed Precision | BFLOAT16 |
| Total Epochs | 2 |
| Optimizer | AdamW |
| Warmup Steps | 100 |
| Weight Decay | 0.1 |

Table 4: Hyperparameters for **D3 mid-training** experiments.

## G  Supervised Finetuning

For supervised fine-tuning, we train models using the Huggingface `accelerate` and DeepSpeed `Zero++` (Wang et al., 2023a) libraries for Pytorch. As in mid-training, we add the special "`<|diff|>`" token to non-mid-trained model vocabularies, and resize embeddings accordingly prior to supervised fine-tuning. Examples from all datasets are reformatted to match the format of examples in D3 prior to fine-tuning.

|  | Llama 3.2 1b & 3b |
| --- | --- |
| Batch Loss Reduction | sum |
| Batch Size | 512 |
| Flash-Attention | true |
| Gradient Clipping | 1 |
| Learning Rate Scheduler | linear |
| Max Learning Rate | 1e-4 |
| Max Sequence Length | 4096 |
| Mixed Precision | BFLOAT16 |
| Total Epochs | 2 |
| Optimizer | AdamW |
| Warmup Ratio | 0.01 |
| Weight Decay | 0.01 |

Table 5: Hyperparameters for **task-targeted supervised fine-tuning**.

# H  More on our Ablation Study (subsection 4.1, Figure 5)

## H.1  Computing Seed Corpus Validation Loss

```
'''
Check if a particular process is running based on its name. If it is not running, start it using the process name as a command. If it is running,
terminate the process and restart it by executing the process name as a command.
The function should output with:
    str: A message indicating the action taken:
    "Process not found. Starting <process_name>."
    "Process found. Restarting <process_name>."
You should write self-contained code starting with:

import subprocess
import psutil
import time
def task_func(process_name: str) -> str:
'''

'''

import subprocess
import psutil
import time
def task_func(process_name: str) -> str:

    # Check if the process is running
    is_running = any([proc for proc in psutil.process_iter() if proc.name() == process_name])

    # If the process is running, terminate it
    if is_running:
        for proc in psutil.process_iter():
            if proc.name() == process_name:
                proc.terminate()
                time.sleep(5)
        subprocess.Popen(process_name)
        return f"Process found. Restarting {process_name}."
    else:
        subprocess.Popen(process_name)
        return f"Process not found. Starting {process_name}."
```

Figure 14: An example of the **prompt-completion formatting that we use to compute the validation fit induced by source code corpuses** before and after the application of our LLM-assisted filtering procedure. Examples are fully formatted Python-executable code, with instructions provided as multi-line comments (bolded text). As indicated in subsection 4.1, prompt tokens are masked from validation loss computation.

## H.2  Synthesis Benchmark Scores of Diff-Action vs. File-State LMs Mid-Trained on D3

In Table 6 and Table 7, we report the numerical scores of diff-sequence vs file-state LMs (Llama 3.2 1b) mid-trained on D3 on HumanEvalSynth and MBPP, respectively. These scores reflect the intermediate mid-training checkpoint with the best test-time scaling curve (i.e. highest best-of-n scores vs. generated tokens across $n$) on the validation set of MBPP*. The full test-time scaling curves for these checkpoints are plotted in Figure 5 (right). We use multiple sampling temperatures, $t \in \{0, 0.1, 0.2, 0.5\}$, to evaluate models.

| | | Scratch | | Completion | |
|---|---|---|---|---|---|
| Size | Diffs? | Pass@1 | Pass@16 | Pass@1 | Pass@16 |
| 1b | ✗ | $5.63 \pm 0.41$ | $14.98 \pm 0.56$ | $15.22 \pm 0.51$ | $37.94 \pm 0.64$ |
| 1b | ✔ | $\mathbf{6.71} \pm 0.73$ | $\mathbf{17.54} \pm 0.57$ | $\mathbf{17.01} \pm 0.52$ | $\mathbf{40.33} \pm 0.65$ |

Table 6: Scores on HumanEvalSynth(+) ("$\pm$" shows standard error over $n = 20$ samples).

| | | Scratch | | Completion | |
|---|---|---|---|---|---|
| Size | Diffs? | Pass@1 | Pass@16 | Pass@1 | Pass@16 |
| 1b | ✗ | $4.64 \pm 0.20$ | $30.04 \pm 0.39$ | $9.29 \pm 0.48$ | $28.62 \pm 0.70$ |
| 1b | ✔ | $\mathbf{10.32} \pm 1.57$ | $\mathbf{33.68} \pm 0.41$ | $\mathbf{10.23} \pm 0.48$ | $\mathbf{33.95} \pm 0.71$ |

Table 7: Scores on MBPP(+) ("$\pm$" shows standard error over $n = 20$ samples).

---

*Note that Figure 4 and Table 2 instead reflect the pass@1 benchmark scores of the *last* D3-mid-trained Llama 3.2 1b checkpoint.

# I  Benchmark Evaluations

We evaluate models on coding benchmarks using instruction-style prompts and no special stop sequences during sampling (i.e. EOS-token termination only).

## I.1  Completion-Style Synthesis Evaluations

As described in subsection 4.1 and shown below in subsection I.3, our benchmark evaluations test LMs on two tasks, synthesis from an empty file ("from scratch") and from a partial human-written solution. Our procedure for preparing the latter task is simple. For each example in a benchmark test set, we: (1) remove a random subset of lines from the body of the provided human-written solution; and (2) check that the resultant code is syntactically correct with the Python code analysis tool `pylint`, re-sampling until this is the case. Preparation of this task is conducted once for each benchmark, so that all models are evaluated on exactly the same set of partial completions.

Examples whose solutions contain a single line of code in the function body are omitted. Furthermore, if the solution to a programming problem contains a docstring, we include this docstring in the partial solution provided to the LM.

## I.2  Examples of Code Synthesis Prompts

```

[start of is_not_prime_solution.py]

[end of is_not_prime_solution.py]


<issue>
Write a Python function `is_not_prime(n)` to solve
the following problem: Identify non-prime numbers.
</issue>


```

```

[start of is_not_prime_solution.py]
  1 import math
  2 def is_not_prime(n):
  3     if n == 1:
  4         return True
  5     return False
  6
[end of is_not_prime_solution.py]


<issue>
Complete the Python function is_not_prime.
</issue>


```

Figure 15: Example **formatted prompts used for from-scratch (left) and completion-style (right) synthesis evaluations** on the HumanEvalSynth and MBPP benchmarks throughout this paper. We use identical prompt-completion formatting in all training and evaluation experiments, including mid-training on D3 as well supervised fine-tuning on all task-curated datasets (subsection 4.2). The examples presented in the figure above reflect a problem from the MBPP test set. Note that for from-scratch-style evaluations, we format MBPP instructions to match the formatting of those in HumanEvalSynth.

## I.3 Example Code Editing Prompt

```

[start of has_close_elements_solution.py]
 1 from typing import List
 2
 3
 4 def has_close_elements(numbers: List[float], threshold: float) -> bool:
 5     """ Check if in given list of numbers, are any two numbers closer to each other than a
 6     given threshold.
 7     >>> has_close_elements([1.0, 2.0, 3.0], 0.5)
 8     False
 9     >>> has_close_elements([1.0, 2.8, 3.0, 4.0, 5.0, 2.0], 0.3)
10     True
11     """
12     for idx, elem in enumerate(numbers):
13         for idx2, elem2 in enumerate(numbers):
14             if idx != idx2:
15                 distance = elem - elem2
16                 if distance < threshold:
17                     return True
18
19     return False
20

[end of has_close_elements_solution.py]


<issue>
Fix bugs in has_close_elements.
</issue>


```

