# OpenReview forum: "D3: A Dataset for Training Code LMs to Act Diff-by-Diff"
_colmweb.org/COLM/2025/Conference — COLM 2025_

### Official Review · Reviewer_9pRV · 2025-05-05

**Rating:** 6
**Confidence:** 3
**Ethics Flag:** 1

**Summary:**

This paper introduces a dataset containing 8 billion tokens of instruction + file-state + file-diff- sequence examples sampled from 850,000 human-written Python source files, to improve LM performance on code synthesis, completion and editing. Experiments show that mid-training LMs like Llama 3.2 1b and 3b on D3 prior to supervised fine-tuning (SFT) on task-curated data improves performance on synthesis & editing tasks. On benchmarks like HumanEval- Synth and HumanEvalFix, D3-trained models show improvements in pass@1 of 3 to 6 points compared to direct SFT.

**Reasons To Accept:**

1. This paper curates a dataset containing 8 billion tokens of instruction + file-state + file-diff- sequence examples to improve LM performance on code synthesis, completion and editing.
2. The paper is well-written, where problem formulation and data curation process are clear and easy to follow.
3. Experiments on several coding benchmarks show D3 improves the performance of small LMs on synthesis & editing tasks.

**Reasons To Reject:**

1. Although authors have submitted the codebase of this paper, there is no dataset released or provided.
2. This paper only focuses on Python, raising questions about its generalizability to other programming languages.

---

> ### Author Response · Authors · 2025-06-01
> **Rebuttal from Authors**
>
> Dear Reviewer,
>
> Thank you for your review. We are delighted that you found our paper to be well-written, and the problem & data formulation formulation to be clear and easy to follow.
>
> > Although authors have submitted the codebase of this paper, there is no dataset released or provided.
>
> We would like to confirm that we are committed to fully open-sourcing the D3 dataset.
>
> > This paper only focuses on Python, raising questions about its generalizability to other programming languages.
>
> All parts of our data preparation pipeline are simple to adapt to other programming languages.
>
> For example, source code filtering with an LLM-Judge is trivial to adapt to another programming language. Furthermore, the LintSeq algorithm that we use for generating synthetic code edit actions throughout the paper can be run on source code written in any programming language, as long as there exists some kind of code linter or verifier for this language.
>
> We focused on Python in this paper due to compute limitations, but we are excited to explore extending D3 to other languages and to repository-level data in future work.
>
> ---
> Thank you once again for your comments. Please let us know if you have any outstanding questions or concerns that could help us increase our submission’s score.

---

> > ### Comment · Reviewer_9pRV · 2025-06-06
> >
> > Thanks for your response! I will keep my rating.

---

### Official Review · Reviewer_zkKL · 2025-05-09

**Rating:** 7
**Confidence:** 5
**Ethics Flag:** 1

**Summary:**

This paper argues for mid-training a Code LLM to edit code. The edit data is structured as the sequence [original code, NL instruction, diff to original code]. The paper contributes a procedure for generating these edits from raw code data, along with a dataset of 8B tokens worth of edits.

The evaluation ablates various design choices and shows that mid-training followed by SFT improves performance slightly for 1B and 3B models on downstream tasks (HumanEval, MBPP, SWE-Bench).

**Questions To Authors:**

1. What are the task-curated SFT datasets? Please let me know if I missed it in the paper, but the dataset does not appear to be explicitly mentioned in Section 4.

2. Caption for Figure 5 says, "Base models are evaluated with the D3 prompt-completion structure (see Figure 2)" Does it make sense ot evaluate the base models with the D3 prompt? Shouldn't they be evaluating using the standard prompt format for HumanEval, MBPP, etc.?

## Notes Taken While Reading

L10: "sampling synthetic file-diff sequences with a code analysis tool". Not
clear what this means at this point in the abstract.

L60: Why is this a point about just validation loss? Does it not also hold for
the test sets / benchmarks?

L84: Shouldn't $\mathcal{T}$ also be indexed by a goal? The actions taken will
depend on the goal, just as the reward does.

L105: The referenced figures are far too terse to understand. But, I'm going on
reading the rest of the section.

L143: OK, I have understood the procedure. Some critique: the sequence of
diffs is entirely additive. It also does not support quite common kinds of
additions, e.g., "add an argument to this function".

Looking back at the prior paper, the delta appears to be that the LintSeq
paper did not have NL instructions, worked by pretraining very small LLMs,
and used a very different dataset.

L176: Some basics about batch size and sequence length would be helpful to see.

Table 2: The "compl." columns. I'm not sure whether this is relevant. It is not
how the benchmark data is typically used.

L268: "Our dataset builds directly upon this work, using the LintSeq algorithm
for refactoring filtered, human-written source code files at pre-training
scales."

I don't think 850K examples is pre-training scale. Actually, that's the original
source -- I think 8B tokens is likely mid-training scale.

**Reasons To Accept:**

1. A nice approach to dataset generation. It's on a small scale, but it is likely to be scalable.
2. Results on 1B and 3B models are likely significant. I think it is quite hard to get +5% Python performance by doing anything to Llama 3.2 1B. But, I have questions about what the baseline is (see below).

**Reasons To Reject:**

1. The delta that this paper presents over *Piterbarg et al. (2024)* is not that large. From what I can tell, the edit synthesis algorithm is not new at all. What is new is that edits are labelled with instructions, there is a larger dataset that is curated with standard techniques, and new experiments.

2. Baseline results are quite unclear, which makes it hard to judge the significance of the experiments.

   See questions below: I cannot determine what the SFT dataset is that the paper uses. This makes the results hard to process. For example, I believe Meta Llama 3.2 Instruct does much better on HumanEval than what this paper reports. So, I don't think the SFT dataset is the Meta SFT dataset.

---

> ### Author Response · Authors · 2025-06-01
> **Rebuttal from Authors**
>
> Dear Reviewer,
>
> Thank you for taking the time to read and review our paper, and for providing your notes!
>
> > Not clear what this means at this point in the abstract.
>
> Thank you for this comment, we will adjust the wording in this sentence.
>
> > Why is this a point about just validation loss? Does it not also hold for the test sets / benchmarks?
>
> Indeed, we can also report test loss. We will include this in our revision.
>
> > Shouldn't T also be indexed by a goal?
>
> The transition function represents the change in state induced by applying an action, i.e. it is a transformation akin to "git apply" and is independent of goal. You may be thinking of the LM policy over the MDP -- the policy indeed depends on the goal.
>
> > Some basics about batch size and sequence length would be helpful to see.
>
> We provided all of our training hyperparameters in Appendix tables 4 and 5.
>
> > I cannot determine what the SFT dataset is that the paper uses
>
> We describe and cite the SFT datasets that we use in our experiments on lines **207 to 212** in Section 4.
>
> Since the D3 dataset is designed to serve as an initialization for any kind of code LM post-training, our experiments treat SFT data as a hyperparameter. We evaluate LMs with and without mid-training on 4 tasks: (a) vanilla synthesis, (b) completion, (c) debugging, and (d) repository-level editing.
>
> Existing open-source SFT datasets are designed for either synthesis or editing tasks, but not both. Thus, to support our multi-task evaluations, our experiments sweep over data mixes from two sources:
>
> (1) **LintSeq-Instruct**, an SFT dataset for vanilla program synthesis with diffs, based on the Python subset of the Magicoder dataset and StarCoder2-Instruct.
>
> (2) **Python-CommitPackFT,** an SFT dataset for program editing. CommitPackFT is sourced and filtered from GitHub commits.
>
> In each experiment, we SFT models on (1), (2), and both (1) & (2). We report the best scores for each benchmark across this sweep. Note that tasks (a), (b), and (c) are in-distribution to our datasets, while (d) tests generalization.
>
> >  The delta that this paper presents over Piterbarg et al. (2024) is not that large
>
> We respectfully disagree. Piterbarg et al showed that for fixed SFT datasets, training LMs to write code with atomic edits can improve vanilla code synthesis.
>
> In contrast, our submission is a dataset paper. Our work extends Piterbarg et al by:
>
> (1) **Merging vanilla code synthesis, code completion, and editing into a single task by training models to natively write code with atomic edits**
>
> (2) **Experimenting with scalable source file preparation for mid-training**, via (a) LLM-judge-based filtering of human-written code and (b) resampling files into error-free edit sequences to get diverse synthesis & completion data
>
> (3) **Testing the interaction between Piterbarg et al.-like SFT and mid-training**
>
> > [The D3 dataset] is curated with standard techniques
>
> Again, we respectfully disagree. To our knowledge, the techniques that we use for curating our dataset have not been explored at-scale for coding LMs in the open-source literature.
>
> For example, though our filtering procedure is indeed inspired by FineMath, this dataset was released in December 2024 and the corresponding paper in February 2025 [5]. Unlike our dataset, FineMath filters natural language data, not code. We believe it is non-trivial that similar, simple LLM-judge filtering is effective for source code.
>
> > Table 2: The "compl." columns. I'm not sure whether this is relevant
>
> The completion-style task that we designed here evaluates an important, practical capability for coding LMs: auto-complete of partially-written code given an instruction. This task is integrated into popular coding assistant tools like Cursor.
>
> Though such evaluations are not standard at the moment, we believe that their omission is problematic and is a factor contributing to the capability gap between open- and closed-source models.
>
> > Does it make sense to evaluate the base models with the D3 prompt?
>
> In this paper, we opt for a stripped-down evaluation procedure in order to decouple the effects of mid-training on D3 from benchmark “hill-climbing” during SFT or at test-time.
>
> Indeed, there are a few choices that we make that contribute to overall lower Llama 3.2 scores:
> - No special stop tokens
> - No reasoning traces
> - A SWE-Bench-like format unifying editing and synthesis. This format is used in all prompts and completions across our training data and evaluations. Examples of it are shown throughout the paper, e.g. in Appendix H (benchmark evaluations).
>
> **Please note that all of our training data (D3 & SFT datasets) reflect these formatting choices** -- as a result, we believe that the comparisons between all pairs of trained models here are fair.
>
> ---
>
> Thank you once again for your very thoughtful review! We would appreciate it if you could let us know if you have any outstanding questions or concerns.
>
> [1] Ben Alla et al. SmolLM2 (2025).

---

> > ### Comment · Reviewer_zkKL · 2025-06-08
> >
> > - Thank you for correcting my oversight about the SFT dataset.
> > - I am also OK with the positioning w.r.t.  Piterbarg et al. (2024) and SmolLM2
> > - I raised my score

---

### Official Review · Reviewer_Wokx · 2025-05-12

**Rating:** 5
**Confidence:** 3
**Ethics Flag:** 1

**Summary:**

The authors introduce D3 - Diverse Data for Diff-by-Diff Coding - as a new dataset for training LLMs for Python code generation. The dataset is constructed by framing the code generation as a decision-making problem with goals, states, and actions defined by the tasks: goal is the description of functionality to be added, state is the current contents of a file, and action is the generation of file diff. The dataset is constructed from Python source files from the Stack, using code analysis tool and LLM to sample file-diff sequences and labels.

**Questions To Authors:**

- Training on D3, how can model perform better through iterative and exploratory code generation like human software development? Can you show empirical results on code generation where the model can improve and refine its code better after training on D3?
- On SWEBench, the results seem quite low even in the Oracle setting. what could be the reason even after training with large-scale dataset like D3?
- The current dataset is for Python code generation. For other languages such as Java, what tools and steps need to be modified to create a similar dataset?

**Reasons To Accept:**

- D3 is a large scale dataset with more than 3.6 million examples
- The dataset contains reasonable technique to augment the data qualities, including synthetic diff generation, LLM-based data filtering and labeling
- D3 has good level of data diversity, from syntactic modification to architectural changes, to source files

**Reasons To Reject:**

- The experiment results are quite limited with most of the results are on simple tasks such as HumanEval and MBPP. However, the motivation of the dataset is to enhance the model with the ability to perform iterative and exploratory code generation like human software development. Except for limited result on SWEBench, it is not clear how the current results reflect the mentioned model capacity. Even on SWEBench, the model is tested on the Oracle version only.
- The creation of the dataset is quite clear. However, many components are controlled by LLM e.g. generating labels. However, there is no study in the paper to justify how accurately these synthetic data are generated by LLMs. As this can affect the data quality, it is important that the authors have a human study (even on a data subset only) to examine the quality of the synthetic components in the dataset.

---

> ### Author Response · Authors · 2025-06-01
> **Rebuttal from Authors**
>
> Dear Reviewer,
>
> We appreciate the time that you have taken to read and provide feedback on our submission. There are a few important points in your review that we would like to respond to and clarify.
>
> > It is important that the authors have a human study
>
> Thank you for this comment. We agree that adding a human study to the paper would strengthen our work. We will conduct such a study before revising our paper.
>
> > the motivation of the dataset is to enhance the model with the ability to perform iterative and exploratory code generation like human software development.
>
> This is not entirely the case. The core motivation of the D3 dataset is to unify code synthesis, completion, and editing into a single problem by training models to *natively* code with diffs or atomic edit actions during mid-training or continued pre-training.
>
> D3 is **not** designed for a hill-climbing on any particular benchmark. Rather, our goal is to develop a source code dataset that can provide a solid initialization for any kind of code LM post-training.
>
> Prior work has shown that for fixed SFT datasets, training LMs to write code with atomic edits (by resampling completions in training data into sequences of synthetic & error-free diffs with LintSeq) can improve vanilla code synthesis [1].
>
> This submission extends prior work by:
>
> (1) **Experimenting with source file preparation for mid-training or late pre-training at scale**, via (a) LLM-judge-based filtering of human-written code and (b) resampling files into error-free edit sequences to get diverse synthesis & completion data
>
> (2) **Studying performance on tasks beyond vanilla code synthesis, like code completion and editing**.
>
> To our knowledge, this paper is the first to test any of (1a), (1b), or (2).
>
> > On SWEBench, the results seem quite low even in the Oracle setting.
>
> Please note that the scores of our 1b and 3b Llama models on the Oracle SWEBench setting are of a comparable order of magnitude to the scaffold-free scores of SWE-Llama 7b from Jiminez et al [1] (~3% pass@1 Resolved, ~66.78% pass@1 Apply). SWE-Llama 7b is more than 2x as large as our 3b model and was also fine-tuned in-task on the train split of SWE-Bench.
>
> In contrast, our evaluations test generalization to SWEBench. Our models are trained on large-scale atomic edit data but are not fine-tuned on any SWE-Bench-like or even multi-file edit data. To improve the clarity of this point, we will add SWE-Llama 7b scores to Table 3.
>
> We chose to evaluate on the Oracle setting and without scaffolding in order to verify that mid-training on our dataset can improve performance independent of scaffold choice. We agree that it is possible to design a SWE-Agent-like scaffold for our models that would substantially improve pass@1, but we believe this to be outside of the scope of this paper.
>
> > results are quite limited with most of the results are on simple tasks such as HumanEval and MBPP
>
> First, we would like to bring your attention to the fact that outside of SWEBench, we also evaluate on both “vanilla” and “completion” variants of HumanEval and MBPP as well as on HumanEvalFix, a popular benchmark for single-file code debugging. Together, these three benchmark variants evaluate our models on three types of coding tasks: synthesis from-scratch, completion, and debugging.
>
> Second, we would like to emphasize that we experiment with 1b and 3b models, due to compute limitations. Models of this scale perform poorly on more challenging competitive coding benchmarks like LiveCodeBench. We could evaluate our models on these benchmarks, but scores would be very low and more highly affected by sampling noise.
>
> Third, we would like to clarify our evaluation philosophy. As stated above, D3 is not designed to hill-climb on a particular benchmark, but rather to provide a good initialization for diverse post-training use-cases.
>
> We believe that the simplest way to evaluate whether the dataset fulfills this objective is by testing the quality of zero-shot and direct generation from LMs, with and without mid-training on D3.
>
> This is why we abstain from using scaffolds, suppress LM reasoning traces, and use no special stop tokens during sampling. These evaluation choices contribute to lower code LM scores, but we believe that they improve the integrity of our results and conclusions.
>
> We would like to assure you that we do believe that scaffolding and reasoning are extremely useful, and we are committed to exploring better post-training for LMs with atomic edit representations in future work.
>
> >  For other languages such as Java, what tools and steps need to be modified to create a similar dataset?
>
> Please see our general response.
>
> ---
>
> Thank you once again for taking the time to review our work. Please let us know if you have any outstanding concerns that stand between us and a recommendation for acceptance.
>
> [1] Jimenez et al. SWE-Bench  (2023).

---

> > ### Comment · Reviewer_Wokx · 2025-06-07
> >
> > Thanks for your response! I appreciate the authors explain the experimental results and the dataset in the rebuttal. I will keep my rating.

---

### Author Response · Authors · 2025-06-01
**General Response from Authors**

We thank all of the reviewers for their insightful and constructive comments. We are glad that you found our data preparation method to be scalable (zkKL) -- yielding data with a good level of diversity (Wokx) -- and our paper to be well-written & clear (9pRV).

We would like to confirm that we will open-source the full D3 dataset (9pRV) and conduct a human study to examine the quality of synthetic labels in D3 (Wokx) during revision.

A few reviewers expressed concerns about the generalizability of our methods to other programming languages (Wokx, 9pRV) and the overall low code LM scores in our evaluations (Wokx, zkKL). Please find a summary of our responses to each of these points below. For more details, please take a look at our individual review rebuttals.

**Generalizability to Other Programming Languages**

All parts of our data preparation pipeline are easy to adapt to other languages.

For example, our data filtering procedure simply relies on an LLM judge to score source files -- the same procedure could be applied without modification to data in other common programming languages like Java or C. For lower resource languages like Lean or CUDA, we would conduct a deeper study evaluating the quality of LLM grades and explore extensions of the current pipeline with techniques like majority voting.

Similarly, the synthetic edit sampling algorithm that we use to transform source files into a source of trajectory-like synthesis or completion data, LintSeq, can be run for code in any programming language. LintSeq only relies on the existence of some kind of linter, verifier, or compiler during samping.

We are excited to explore extending D3 to other programming languages in future work.

**Low Code LM Scores in Evaluations**

The core motivation of the D3 dataset is to unify code synthesis, completion, and editing into a single problem by training models to *natively* code with diffs or atomic edit actions during mid-training or continued pre-training. D3 is not designed for hill-climbing on any particular benchmark, but rather to yield a solid initialization for any kind of code LM post-training.

As a result, in this paper, we opted for stripped-down and simple evaluation procedures, with the view that this would improve the rigour of our conclusions about the effects of mid-training. To that end, we: (1) did not use any stop tokens besides EOS during sampling; (2) evaluated zero-shot, direct prediction (no reasoning); and (3) did not use any scaffolding in evaluations on SWE-Bench. These choices lowered code LM scores across benchmarks.

We intend to study hill-climbing with atomic edit representations -- for example, by training models to reason before generating atomic edits -- in future work.

---

### Decision · Program_Chairs · 2025-07-08

**Decision:**

Accept

**Comment:**

The papaer introduces D3, a large dataset consisting of 8B tokens from 850K Python files. The dataset frames code generation as sequential edits (file-diffs) annotated with goals, constructed by filtering and augmenting code from The Stack using a code analysis tool and labeling with LLM-generated rationales. Mid-training language models (Llama 1b and 3b) on D3, prior to SFT, improves model performance in code synthesis, completion, and editing tasks.

The reviewers praise the approach used to produce the data, its diversity, an the clear presentation. Concerns are mainly about limited evaluation, generalization beyond Python, and evidence justifying the quality of synthetic labels, most of which have been addressed in the rebuttal.

This paper is ready for COLM as is, but I encourage the authors to add the human evaluation experiment that R Wokx suggested in the revision.